# Sleeping sickness is a circadian disorder

Filipa Rijo-Ferreira [1,2,3,4], Tânia Carvalho[2], Cristina Afonso[5], Margarida Sanches-Vaz[2], Rui M Costa[5], Luísa M. Figueiredo [2] & Joseph S. Takahashi [3,4]

Sleeping sickness is a fatal disease caused by *Trypanosoma brucei*, a unicellular parasite that lives in the bloodstream and interstitial spaces of peripheral tissues and the brain. Patients have altered sleep/wake cycles, body temperature, and endocrine profiles, but the underlying causes are unknown. Here, we show that the robust circadian rhythms of mice become phase advanced upon infection, with abnormal activity occurring during the rest phase. This advanced phase is caused by shortening of the circadian period both at the behavioral level as well as at the tissue and cell level. Period shortening is *T. brucei* specific and independent of the host immune response, as co-culturing parasites with explants or fibroblasts also shortens the clock period, whereas malaria infection does not. We propose that *T. brucei* causes an advanced circadian rhythm disorder, previously associated only with mutations in clock genes, which leads to changes in the timing of sleep.

[1] Graduate Program in Areas of Basic and Applied Biology, Instituto de Ciências Biomédicas Abel Salazar, Universidade do Porto, 4099-002 Porto, Portugal. [2] Instituto de Medicina Molecular, Faculdade de Medicina, Universidade de Lisboa, 1649-028 Lisboa, Portugal. [3] Department of Neuroscience, University of Texas Southwestern Medical Center, Dallas, TX 75390-9111, USA. [4] Howard Hughes Medical Institute, University of Texas Southwestern Medical Center, Dallas, TX 75390-9111, USA. [5] Champalimaud Neuroscience Programme, Champalimaud Centre for the Unknown, 1400-038 Lisbon, Portugal. Luísa M. Figueiredo and Joseph S. Takahashi jointly supervised this work. Correspondence and requests for materials should be addressed to L.M.F. (email: lmf@medicina.ulisboa.pt) or to J.S.T. (email: joseph.takahashi@utsouthwestern.edu)

Sleeping sickness is one of the major neglected tropical diseases, with an incidence of 10,000 new cases per year, threatening more than 60 million people in sub-Saharan Africa[1,2]. Sleeping sickness is caused by the unicellular and extracellular parasite Trypanosoma brucei, which can be found in the bloodstream and interstitial spaces of the adipose tissue and skin, but eventually invades the brain, leading to coma and death if untreated[1,3–5]. Although patients experience a variety of symptoms[6,7], the hallmark of sleeping sickness is the disruption of the sleep pattern. Patients experience somnolence during the day and insomnia at night, but with total time spent sleeping similar to healthy individuals[8–11]. The fact that the sleep/wake cycle disruption in patients reverts to normal upon treatment[9], and that autopsies show that patients who die of sleeping sickness lack neuro-degeneration[12,13], suggests that the presence of parasites, rather than neuronal death, is the cause of these symptoms. This curious daily sleep/wake cycle disruption, together with changes in body temperature regulation and disrupted timing of endocrine secretion[7,8,12,14], indicates that sleeping sickness may be a circadian clock rhythm disorder.

Here, we report that T. brucei causes a circadian rhythm disorder in mice, recapitulating the circadian behavior and body temperature changes in humans. Mechanistically, we show that T. brucei infection shortens the period of the mouse circadian activity rhythm at the organismal level as well as at the cellular and molecular level. This acceleration of the host clock is not observed with other infections and can be reproduced in vitro, indicating that it is probably caused by the direct interaction of cells with the parasite or with a parasite molecule.

## Results

**Infected mice are active during the resting phase.** In mammals, neurons within the suprachiasmatic nucleus (SCN) of the hypothalamus orchestrate diverse circadian physiological functions including the sleep/wake cycle, core body temperature, and metabolism[15]. Given that the patients experience somnolence during the day and insomnia at night[8–11], we set out to test whether T. brucei infection affected circadian behavior. We measured the locomotor activity of the infected mice using running-wheel activity, which is a robust and reliable SCN-driven circadian clock output[16]. Mice were individually housed and running-wheel activity monitored with an automated system. Arising from the rotation of the Earth, light cycles are the most important daily environmental cue to which the circadian system can synchronize or entrain. To ensure that the internal circadian clock of mice prior to infection was normal, mice were first entrained to light/dark cycles (LD) for seven days and then housed in constant darkness (DD) for 10 days[16]. As expected for nocturnal animals, prior to infection, every mouse had a normal circadian rhythm and was active during the nighttime in LD, and this rhythmic behavior persisted in DD with a shorter period (approximately 23.7 h, known as "free running period"). Mice were then injected on day 0 either with T. brucei parasites or with vehicle solution (culture medium) (Fig. 1a). We observed that infected mice were four-fold less active than control mice (Fig. 1c, d), in particular in the periods following high parasitemia, when anemia is ~50–70% (6–10 days post-infection, Supplementary Fig. 1a, b). This sickness-like activity reduction is likely due to exacerbated inflammation as a result of high parasitemia, as it has been shown that injection of pro-inflammatory cytokines reduces running-wheel activity[17]. It is interesting to note that although infected animals always start running at the beginning of nighttime, the time spent running changes over days, leading to modulatory 'waves of activity' (Fig. 1c). This is curious since this infection is known for its waves of parasitemia (Supplementary Fig. 1a).

In humans, T. brucei causes relatively low and often undetectable parasitemia. In order to mimic this in mice, animals were treated around day 21 post-infection with suramin (20 mg/kg), a drug that does not cross the blood brain barrier and thus only kills parasites in the periphery[18]. As a result, animals do not develop overt disease and survival is prolonged with undetectable parasitemia (Fig. 1b). When we treated mice with suramin, we observed that infected mice still run 1.5-fold less than control mice (Fig. 1e, f). But strikingly, infected mice had abnormal levels of activity during the daytime (~15% of activity occurs during daytime after day 50, as compared to ~1% of activity in control mice; Fig. 1e, g and Supplementary Fig. 1d). In particular, similar to the circadian activity disruption observed in patients[19], four out of 14 mice ran more than 25% of their daily activity during the rest period after day 50 post-infection (Fig. 1g). This daytime activity increase is maintained even when considering the absolute number of wheel revolutions during the daytime, when infected mice run 4.5-fold more than controls (Supplementary Fig. 1e). This daytime activity is very unusual since the inhibitory masking effects of light on running-wheel behavior are generally very strong[20,21].

**Highest core temperature shifted from nighttime to daytime.** Since the SCN neurons drive not only activity rhythms but also circadian body temperature fluctuations[22], we assessed the effects of T. brucei infection on temperature by measuring core body temperature with an implanted temperature sensor[22]. As expected, in control animals we observed a circadian oscillation of high temperature during the night and low during the day. In infected animals, we observed an initial fever-like peak that lasted two days matching the first peak of parasitemia (day 4–6 post-infection), followed by a normal temperature oscillation (Fig. 1h, i). However, around 60 days post-infection, the highest core temperature shifted from the night to daytime, resembling the shift we observed in the wheel-running activity (Fig. 1h, i). This implanted sensor system also measures general cage activity, which confirmed that at later stages in the infection, general cage activity was increased during the rest (daytime) phase (Supplementary Fig. 1f).

Interestingly, beginning soon after infection, the circadian amplitude of the temperature rhythm was reduced in infected mice (Fig. 1j). Additionally, infected mice showed a hypothermic period later in the infection, below the minimum of normal physiological range of 36–38.5 °C[22], with lower temperature dropping below 34 °C (Fig. 1k). Thus, both circadian activity rhythms and body temperature rhythms are affected in a parallel manner by T. brucei infection.

**T. brucei infection affects both circadian sleep and feeding.** Sleeping sickness is characterized by increased sleep during the daytime but overall patients spend the same amount of time sleeping as healthy individuals[9]. Using an automated video analysis system (CleverSys), we analyzed feeding and sleeping of control and infected mice at days 20, 65, 85, and 95 post-infection. Later in the infection (days 85 and 95), infected mice showed a disrupted feeding pattern and spent more time in a sleep-like state during the active phase than control animals, even though they did not rest more over the 24-h day (Supplementary Fig. 2). This is consistent with a shift in the circadian distribution of sleep that is concordant with the change in the phase distribution of circadian activity and temperature rhythms.

Together these results show that mice infected by T. brucei exhibit a similar disruption of circadian activity, sleep, and daily temperature as described in humans, thus validating this as a model for human sleeping sickness.

***T. brucei* infection shortens the circadian activity period**. To determine whether *T. brucei* interferes with the mouse circadian rhythm, we repeated the same experiments but with mice maintained in DD, when the internal circadian clock controls the timing of rhythmic locomotor behavior. In the absence of light as a temporal cue, the circadian clock imposes a stable wheel-

running activity with a period of ~23.7 h, with more activity during the nighttime (Fig. 2a–d)[23]. We infected mice in the dark and let the infection progress in DD. We observed that running-wheel activity period was ~30 min shorter in infected mice than in control mice (Fig. 2a–d). As a result of this period shortening, after 20 days post-infection, infected mice start to run 12 h earlier

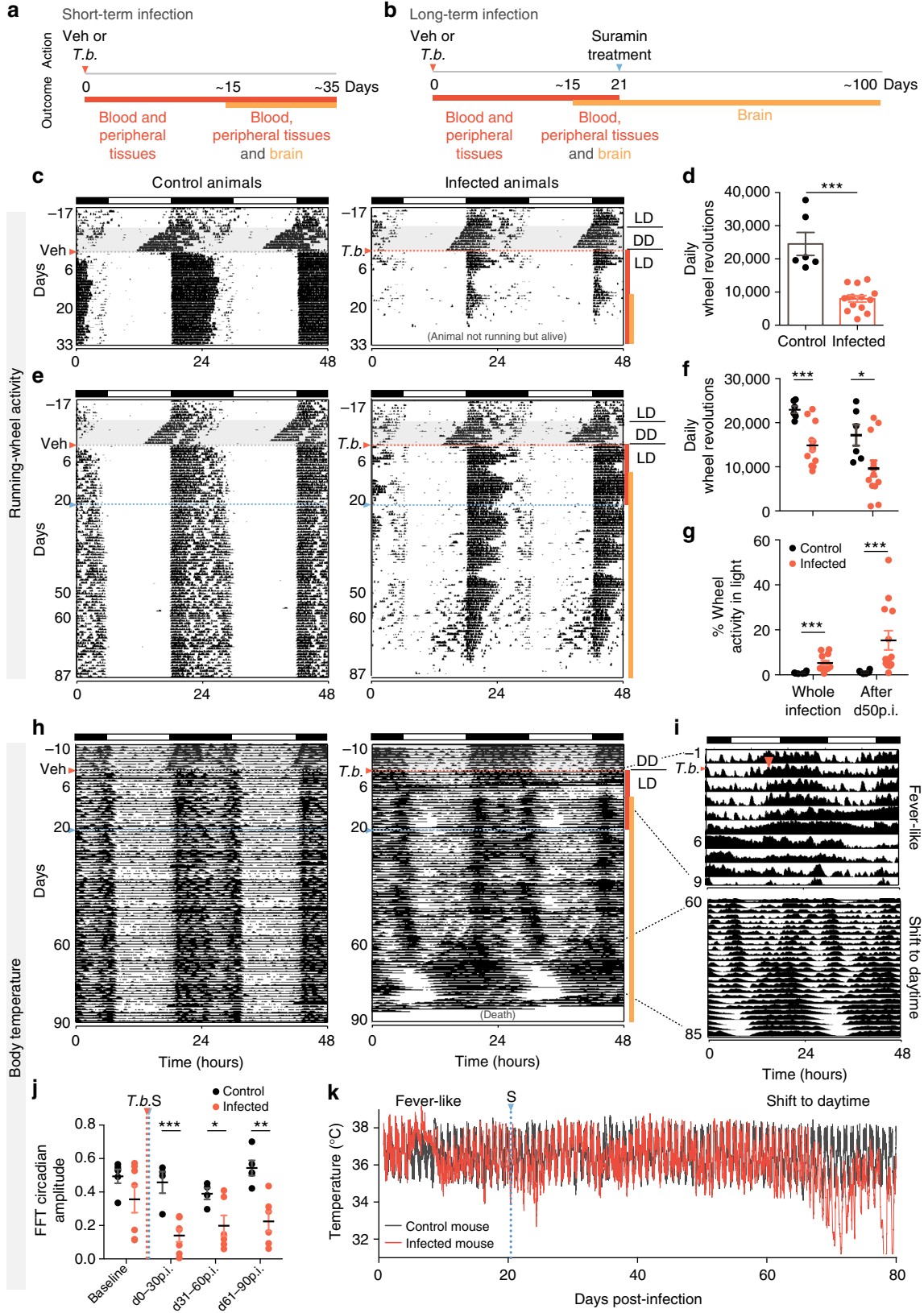

than control mice. Unlike control mice, whose period is stable over time, infected mice showed a progressively shorter period (Fig. 2b–d and Supplementary Fig. 1g): from ~23.5 h in the first 10 days to ~23.1 h between days 70 and 80. Interestingly, this shorter period was detected immediately in the first 10 days post-infection, when very few or no parasites can be detected in the brain (Supplementary Fig. 1c)[24,25]. These results demonstrate that *T. brucei* shortens circadian period in vivo and suggest that early changes in circadian behavior may not be exclusively due to the presence of parasites in the brain, but rather a consequence of a peripheral signal, such as a metabolite or hormone that feedbacks onto the master clock.

**Infected mice have advanced circadian rhythms**. Infected mice showed increased activity during resting phase after 50 days (in LD cycles) and have shorter circadian period than controls when maintained in DD (Figs. 1 and 2). To determine whether the increased daytime activity in LD was due to an underlying phase advance of the circadian rhythms, we transferred mice to DD for 15 days starting on day 60, followed by a period of LD to test ability of re-entrainment and a second dark period. As expected, whenever control animals were released into DD, they maintained their normal free running period (~23.7 h) with activity onsets occurring near the beginning of the dark period at Circadian Time 12 (CT12; Fig. 2e, f). When transferred back to LD12:12, control mice were able to re-entrain rapidly and ran exclusively during the nighttime (Fig. 2e). In contrast, most infected mice showed a phase advance when released into DD with activity onsets occurring before lights out (average phase: CT12.35 in controls and CT11.32 in infected mice, Fig. 2f). Many infected mice could not immediately re-entrain when returned back to LD12:12 (Fig. 2g) and some mice failed to entrain after the second DD episode (Fig. 2e bottom right panel). These results highlight once more that the circadian clock of infected mice is disrupted since these mice show (i) a phase advance when released into DD, (ii) impaired re-entrainment to LD, and (iii) period shortening; all characteristics that are absent in healthy uninfected mice.

**Ex vivo highly parasitized tissues show shorter period**. Within the neurons of the SCN and virtually every cell of our body, there is a circadian transcription–translation feedback loop comprised of a core set of genes: the activators *Clock* and *Bmal1* and their repressors encoded by *Per1/Per2* and *Cry1/Cry2*[26]. This loop takes 24 h to be completed, leading to the rhythmic expression of multiple downstream genes that will then impact a variety of cellular pathways. Since both behavior and core temperature are regulated by clock genes[15], we investigated whether the advanced circadian rhythm could be explained by changes in the expression

of core clock genes. For this, we infected *Per2^{tm1Jt}* mice that express a PERIOD2::LUCIFERASE (PER2::LUC) fusion reporter protein[27] and collected multiple organs to measure circadian PER2 expression ex vivo. Peripheral organs of control mice showed rhythmic oscillations of PER2 expression (Fig. 3a, b). When mice were sacrificed six days post-infection, no changes in period were detected, but interestingly the phase of the adipose tissue (AT) depots was advanced (Fig. 3c). To test whether this change in PER2 expression was stronger once more parasites have infiltrated the organs[4], we tested the expression of PER2::LUC in those same organs on day 20. At 20 days post-infection, most infected organs kept oscillating with normal circadian parameters of phase, damping and period similar to that of the controls (Fig. 3a–c and Supplementary Fig. 3a, see Methods section). Adipose tissue, however, which has the highest parasite load[4], had ~2 h shorter period (Fig. 3a, b and Supplementary Fig. 3b). This tissue also showed a phase advance as observed on day 6 and notably remained cycling, with damping remaining unaffected. This was observed in both gonadal and perirenal adipose tissue depots (Fig. 3b). The period shortening observed on day 20 and phase advance for both days 6 and 20 in adipose tissue suggests that even though on day 6 the presence of parasites was not enough to shorten the period, these depots were already phase advanced relative to controls, and as the parasite load increased, a shorter period was detectable, even ex vivo.

**Eliminating parasites from tissues reverses period shortening**. To test whether the shorter period of infected adipose tissue could be reversed by eliminating parasites, we treated animals with suramin and assessed PER2::LUC circadian rhythms at day 60. When parasites were eliminated from peripheral organs, the period of the adipose tissue was rescued and became similar to control organs (Fig. 3d, e), suggesting that the shorter period was a consequence of high parasite load. Suramin cannot eliminate parasites from the central nervous system and remarkably, on day 60 the period of the SCN of infected mice was 30 min shorter than control SCNs (Fig. 3d–f and Supplementary Fig. 3b). This is particularly interesting because it was approximately on day 60 in LD conditions that we observed phase advances of the circadian activity and body temperature of infected mice (Fig. 1e, g). Together these data suggest that cumulative parasite load in the tissues leads to changes in the period of circadian clock, which can be reversed upon treatment with suramin for parasite elimination.

**Plasmodium infection does not change host circadian period**. The molecular clock is driven by a transcription–translation feedback loop. To test to what extent the clock was affected, we assessed the expression of other clock genes by infecting mice

**Fig. 1** *T. brucei* infection disrupts circadian activity and body temperature. **a, b** Schematic representation of infection progression when animals are (**a**) infected or (**b**) infected and treated with the anti-trypanosome drug suramin. **c** Representative actograms of daily wheel-running activity of control and infected mice in light–dark cycles. The record is double-plotted so that 48 h are shown for each horizontal trace. Black daily histograms represent locomotor activity. For the first seven days, the animals were housed in an LD12:12 h cycle, denoted by the bar above the record. The animals were transferred to DD conditions for 10 days, as indicated by the horizontal line to the right of the record and shaded gray. After these LD and DD intervals, animals were transferred back into LD and either infected or injected with vehicle. The red and orange lines near the actogram refers to the infection progression represented in panel **a**. **d** Total daily activity levels of control ($n = 6$) and infected ($n = 14$) mice. **e** Representative actograms of daily wheel-running activity of animals treated with suramin. The red and orange lines near the actogram refers to the infection progression represented in panel **b**. **f** Total daily and **g** relative rest-period activity levels of control ($n = 6$) and infected ($n = 14$) mice considering the whole infection time and only after day 50 post-infection. Error bars show mean ± s.e.m; *$p < 0.05$, **$p < 0.01$, ***$p < 0.001$ tested with Mann–Whitney test in panels **d**, **f**, and **g**. **h** Representative plots of core body temperature of control and infected mice (suramin-treated) in light–dark cycles. **i** Zoom-in temperature plots showing the fever-like period after infection and the later period of temperature shift to daytime. **j** Amplitude of circadian temperature oscillation in control ($n = 5$) and infected ($n = 7$) mice. Error bars show mean ± s.e.m.; *adjusted $p < 0.05$, **$p < 0.01$, ***$p < 0.001$, FDR method Benjamini–Hochberg. **k** Absolute temperature measurement for a representative control and an infected mouse. *T.b.* orange dotted line represents infection with trypanosomes and S, blue dotted line represents suramin treatment

kept in darkness and, on day 20 post-infection, collecting and extracting RNA from the organs[21]. We measured the mRNA expression of genes involved in the molecular clock: transcriptional activator *Bmal1*, transcriptional repressor *Per1*, and an immediate downstream target of the clock D site of albumin promoter (albumin D-box) binding protein (*Dbp*). As expected,

even after 20 days in DD, clock gene expression in control mice cycled in liver, adipose tissue, and hypothalamic area of the brain, where the SCN is located (Fig. 4a and Supplementary Fig. 4). In control mice, we observed the expected daily oscillations of *Bmal1*, with its maximum expression at circadian time 24 h (CT24). BMAL1 protein heterodimerizes with CLOCK and

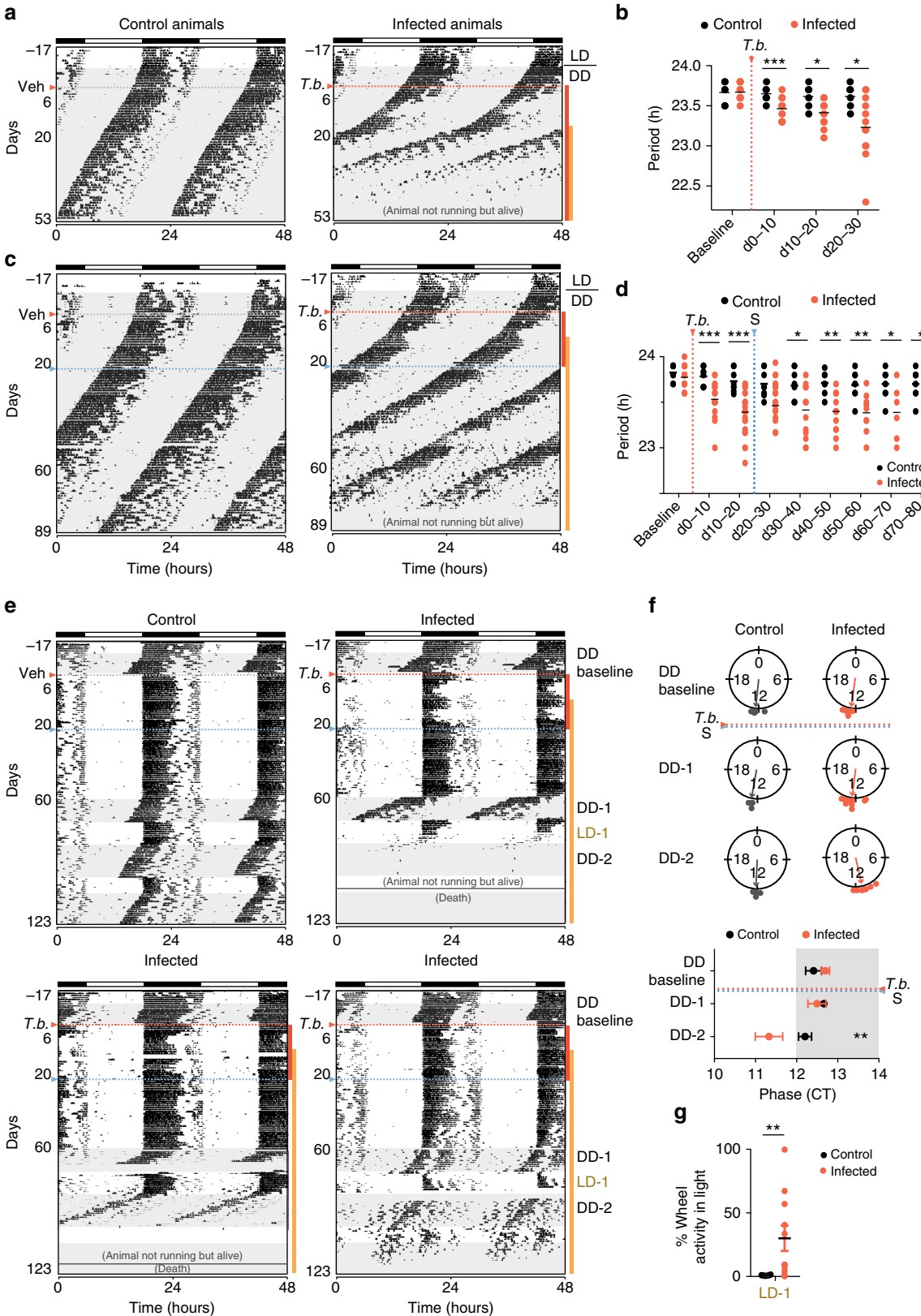

activates transcription of *Per1* and *Dbp*, whose maximal expression occurs at CT8–12. In sharp contrast, the circadian transcript profile of clock genes is highly disrupted when mice are infected with *T. brucei* (Fig. 4a and Supplementary Fig. 4) with an abnormal phase of *Per1* expression in the liver, a lower amplitude of the *Dbp* rhythm in the hypothalamus, a dramatic downregulation of mRNA levels of most clock genes, and oscillations becoming undetectable in the adipose tissue.

To test whether the disruption of clock gene circadian expression is a specific effect of *T. brucei* infection and not due to an immune response to a parasitic infection, we infected mice with a rodent malaria parasite, *Plasmodium chabaudi*. Both parasite species cause chronic infections in the mouse, with comparable immune responses and similar tissue tropism patterns (Fig. 4c, d)[28]. We infected mice with *P. chabaudi* and collected organs every 4 h to measure transcript levels of the same clock genes by qPCR. We observed that the mRNA expression profile of clock genes in the liver was similar in both control and Plasmodium-infected mice. However, in the adipose tissue, expression of clock genes was in general lower in Plasmodium-infected mice (Fig. 4b). This is likely due to inflammation, since inflammation upregulates TNF and it has been previously demonstrated that TNF impairs expression of clock genes in synchronized NIH 3T3 fibroblasts[17]. We confirmed that infections led to upregulation of pro-inflammatory cytokines (TNF and IL-1) and downregulation of the IL-10 anti-inflammatory cytokine (Fig. 4c) and found similar inflammatory cell infiltrates, in liver and adipose tissue, for both Trypanosoma- and Plasmodium-infected mice (Fig. 4d).

Importantly, in the same NIH 3T3 fibroblast study, although inflammation led to downregulation of clock genes, the circadian period of oscillations was unchanged[17]. We repeated the running-wheel activity recordings with mice infected with *P. chabaudi*. Similar to what we observed with Trypanosoma-infected mice (Fig. 1c–g), malaria infection induced a sickness-like behavior, after the first peak of parasitemia, with mice running less than controls (Fig. 5a, c). However, unlike trypanosome infection, malaria-infected mice did not increase their activity during daytime (Fig. 5c), suggesting that these mice have normal circadian rhythms. To further confirm this, we recorded malaria-infected mice in DD. Once again there was a decrease in activity, but the period of the circadian clock remained the same: both control and infected mice showed a running-wheel activity period of 23.69 h ± 0.06 s. d. (Fig. 5b, d). Similarly, the molecular clock remained normal upon malaria infection. None of the tested organs of *P. chabaudi*-infected PER2::LUC mice showed significantly different period than the control mice (Fig. 5e), indicating that, unlike *T. brucei*, *P. chabaudi* infection does not cause detectable period changes.

**T. brucei is sufficient to shorten period of fibroblasts**. Together our data reveal two different effects of *T. brucei* infection: downregulation of clock gene expression (common to other infections, inflammation-driven) and period shortening (probably trypanosome specific). To exclude the contribution of inflammation further, we co-cultured healthy explants of SCN, lung, gonadal adipose tissue, and epididymis of PER2::LUC mice with $10^5$ *T. brucei* parasites. Luminescence was recorded for five days and circadian parameters estimated as above. In the presence of parasites, PER2 expression period tended to be shorter, being significantly shorter in the adipose tissue explant (Fig. 6a). This experiment suggests that the period shortening is possibly due to a secreted molecule from the parasite, since the explant is separated from the parasite culture by a 0.4-µm pore membrane. The absence of a significant difference in PER2 expression period in the SCN, epididymis, and lung may be due to a lower ratio of parasites to host cells, since these tissues are composed of a higher number of cells per milligram. Unfortunately, there are technical limitations in order to culture higher numbers of parasites because in high density cultures ($1–3 \times 10^6$/mL), *T. brucei* differentiate into non-replicative stumpy forms and die soon after[29]. Unlike what is observed in normal culture conditions where starting a culture at $5 \times 10^4$/mL leads to parasite death on day 4, parasites were alive until day 5 in our recording conditions, with the population being ~80% composed by stumpy forms on this day (Supplementary Fig. 5a, b). Since it was not possible to use higher numbers of viable parasites, we took an alternative approach. We co-cultured low density *T. brucei* cultures with primary and immortalized fibroblasts of mouse PER2::LUC and observed that, in the presence of *T. brucei*, PER2 circadian period in both fibroblast types was 30 min shorter (Fig. 6b, c and Supplementary Fig. 5c). Taken together, these data demonstrate that an inflammatory response via the immune system is not responsible for the shortening of clock gene period and that parasite presence is sufficient to induce these period changes.

Given the above in vitro results (parasite sufficiency for period modifications) and that all life cycle stages in *T. brucei* are extracellular, we investigated the existence of a parasite structural protein/lipid or secreted factor(s) that could induce changes in the period of the molecular clock in mammalian cells. We cultured fibroblasts with parasite lysates, with the very abundant and cleaved trypanosome variant surface glycoprotein (sVSG)[30,31] or conditioned media. Parasite lysates did not shorten the period of the fibroblasts' clock (Supplementary Fig. 5d), which is not surprising since we observed a shortening of the period even in the presence of a separating membrane (Fig. 6a). Adding sVSG to the media shortened the period of the lung circadian clock, but it did not shorten the period of the adipose tissue or fibroblasts (Supplementary Fig. 5e–g). Finally, although variability in the

---

**Fig. 2** Trypanosoma-infected mice show shorter circadian period. **a** Representative actograms of daily wheel-running activity of control and infected mice in constant darkness (DD). All running-wheel experiments involve seven days on LD12:12 followed by 10 days in DD to confirm that all animals have a normal circadian rhythm, after which animals are either infected or injected with vehicle in the dark. Horizontal black and white bars at the top of each actogram represent lights off and on for the initial acclimatization period, respectively. **b** Period of running-wheel activity of control ($n = 6$) and infected ($n = 14$) mice. **c** Representative actograms of daily wheel-running activity of control and infected mice in constant dark. Animals were treated with suramin (20 mg/kg) i. p. on day 21 post-infection (blue line). **d** Period of control ($n = 8$) and infected ($n = 27$) mice; *adjusted $p < 0.05$, **$p < 0.01$, ***$p < 0.001$, FDR method Benjamini–Hochberg, for panels **b** and **d**. **e** Representative actograms of daily wheel-running activity of a control and three infected animals when challenged with consecutive dark to light period transitions. DD-1 first period in DD started on day 60 post-infection, LD-1 when mice were transferred to LD, and DD-2 when transferred to a new DD period. **f** Circular phase plots of activity onset for the dark periods represented on the top panel. A circle represents a 24-h clock, and activity onset phases of individual mice were calculated as angles and plotted as colored symbols outside the circle. The direction of the vector indicates mean phase angle. Phase data are replotted on bottom panel as Circadian Time (CT). Shaded area represents subjective night; **adjusted $p < 0.01$, FDR method Benjamini–Hochberg. **g** Relative daytime activity in the light period after the dark period 1; **$p < 0.01$, tested with Mann–Whitney test. Orange dotted line represents *T. brucei* infection start and blue dotted line the suramin treatment

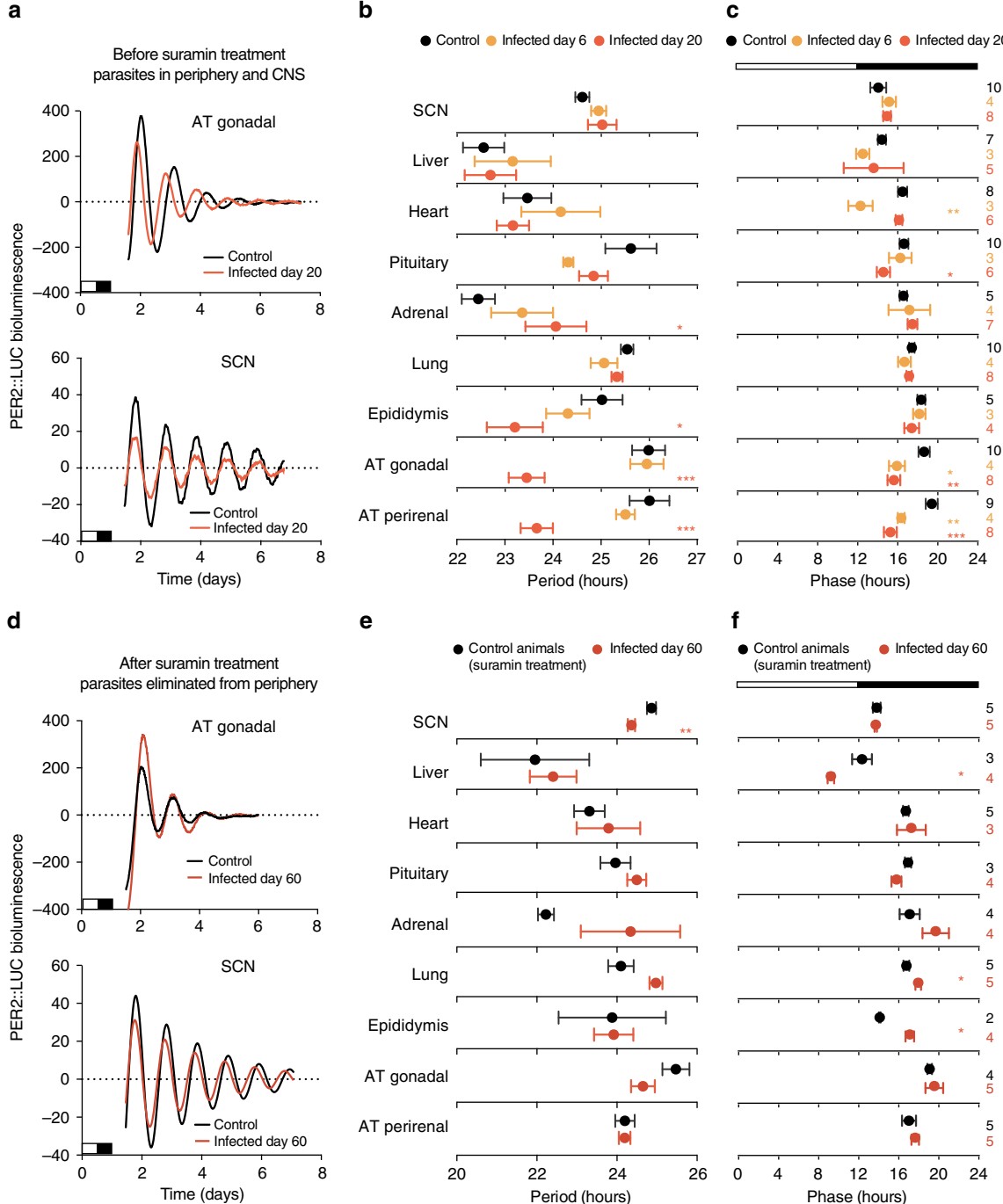

**Fig. 3** Period of PER2::LUC explants becomes shorter when high number of parasites are present. **a** Representative normalized records of bioluminescence reporting PER2 circadian expression from gonadal adipose tissue and SCN on day 20 post-infection/vehicle. White and black box on the x-axis represent the light/dark cycle of mice before explants. **b**, **c** Period and phase plots of various tissues harvested from control (black) and infected mice (day 6 yellow, day 20 orange) mice. **d** Representative records of bioluminescence reporting of circadian expression from gonadal adipose tissue and SCN on day 60 post-infection/vehicle in animals treated with suramin (20 mg/kg). **e**, **f** Period and phase plots of various tissues harvested from control (black) and infected mice (day 60 dark orange) mice. Tissues were prepared from mice in LD. Shown are seven days of continuous recording after explant preparation. The sample size is indicated on the right. Shown are mean period ± s.d. for **b**, **c** and **e**, **f** panels; * adjusted $p < 0.05$, ** adjusted $p < 0.01$, *** $p < 0.001$, FDR method Benjamini–Hochberg

period of fibroblasts increased when cultured with *T. brucei*-conditioned media, the period length remained the same (Supplementary Fig. 5h), suggesting that this molecule may be labile. Overall our data strongly suggest that the presence of *T. brucei* parasites, and not a systemic immune response, is responsible for the changes in period of the circadian clock of the host.

## Discussion

As the name implies, sleeping sickness is considered a sleep disorder. However, the timing of sleep[7,32], activity[19], and endocrine secretion[8] observed in patients suggests a circadian aspect to this disease. Here, we validate a mouse model that recapitulates these observations made in patients and provides evidence that sleeping sickness is a circadian disorder.

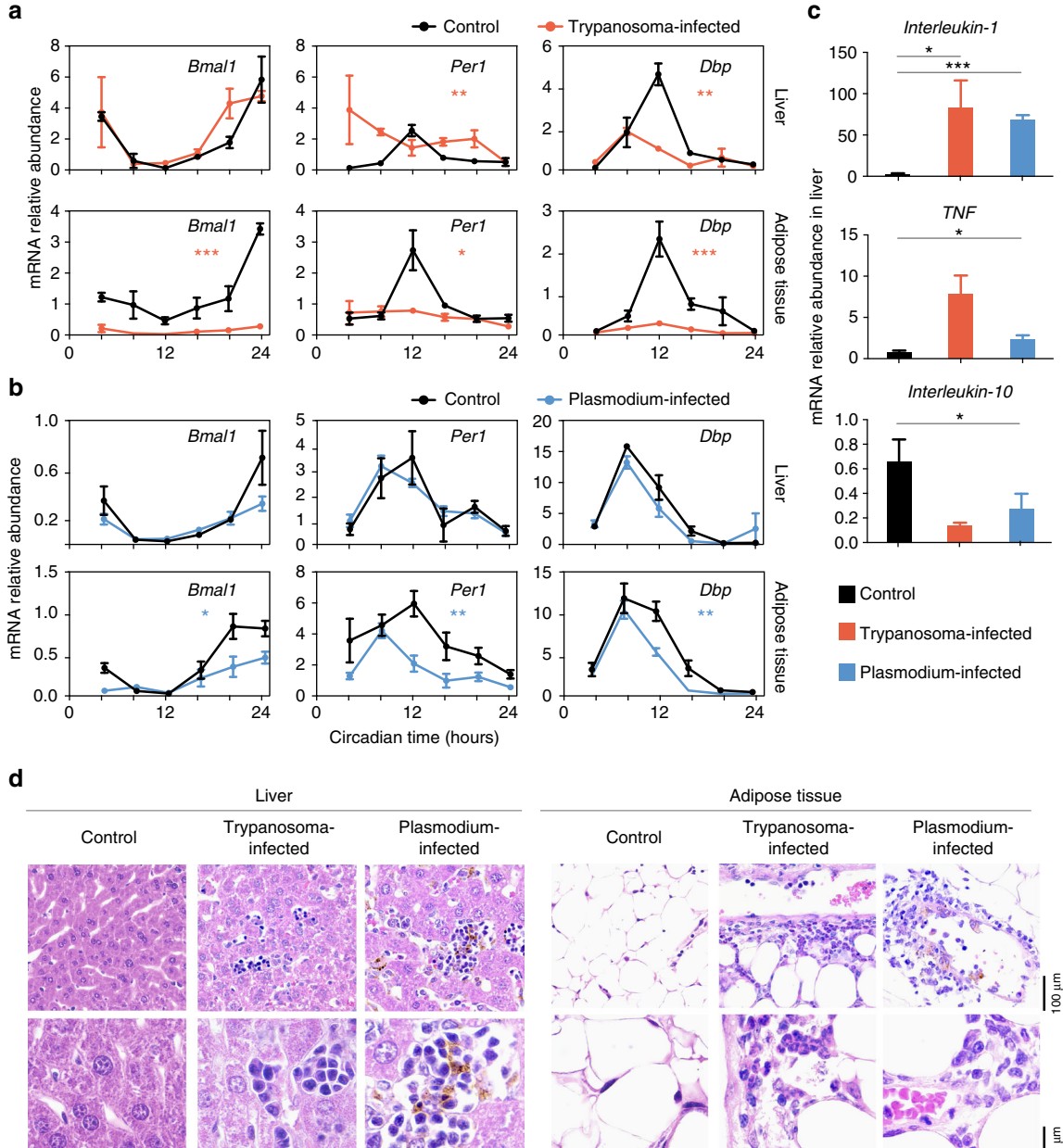

**Fig. 4** Shorter period in circadian clock genes is *T. brucei* specific. **a** Real-time qPCR analysis of clock gene expression in control (black) and Trypanosoma-infected (orange) mice. **b** Real-time qPCR analysis of clock gene expression in control (black) and Plasmodium-infected (blue) mice. Two organs are shown, liver and gonadal adipose tissue. Error bars represent ± s.e.m. for each time point from three mice. Two-way ANOVA comparing differences between groups shows significant statistical differences between control and Trypanosoma-infected mice, * $p < 0.05$, ** $p < 0.01$, *** $p < 0.001$. For visualization purposes, points are united with a line. **c** Real-time qPCR of cytokine mRNA expression in liver; **adjusted $p < 0.01$, ***$p < 0.001$, FDR method Benjamini–Hochberg. **d** Representative microphotographs of liver and gonadal adipose tissue from non-infected, Trypanosoma-infected and Plasmodium-infected mice; depicted are the inflammatory cell infiltrates (arrow) seen in both tissues, for both infection models; $n = 5$ per condition. Hematoxylin and Eosin. Original magnification ×40 (upper panel) and ×100 (lower panel)

When the circadian behavior of infected mice was recorded in the dark, we detected a short circadian period soon after infection before parasites accumulate in large numbers in the brain. Running behavior is controlled by the master clock in the SCN[16]; however, these observations suggest that the presence of high numbers of *T. brucei* in the brain is not necessary for the period shortening of the master clock. Instead, it raises the possibility that a peripheral molecule(s), metabolite or hormone, from the host or directly produced by the parasite, is released from the periphery and feeds back to the master clock. This observation is consistent with a study in Uganda, in which 57% of early blood-

stage (when parasites have not yet reached the central nervous system) *T. brucei rhodesiense*-infected patients already experienced somnolence[33]. Similar observations were made in *T. brucei gambiense*[6,34].

Despite the 2-h period shortening observed in the periphery, ex vivo analysis of PER2::LUC expression from SCN of *T. brucei*-infected mice showed no differences relative to controls on days 6 and 20 post-infection. This suggests that when isolated from the periphery in culture, the SCN clock in the brain appears unaffected. On the other hand, 60 days post-infection, the period of PER2::LUC expression in the SCN was shorter than in controls.

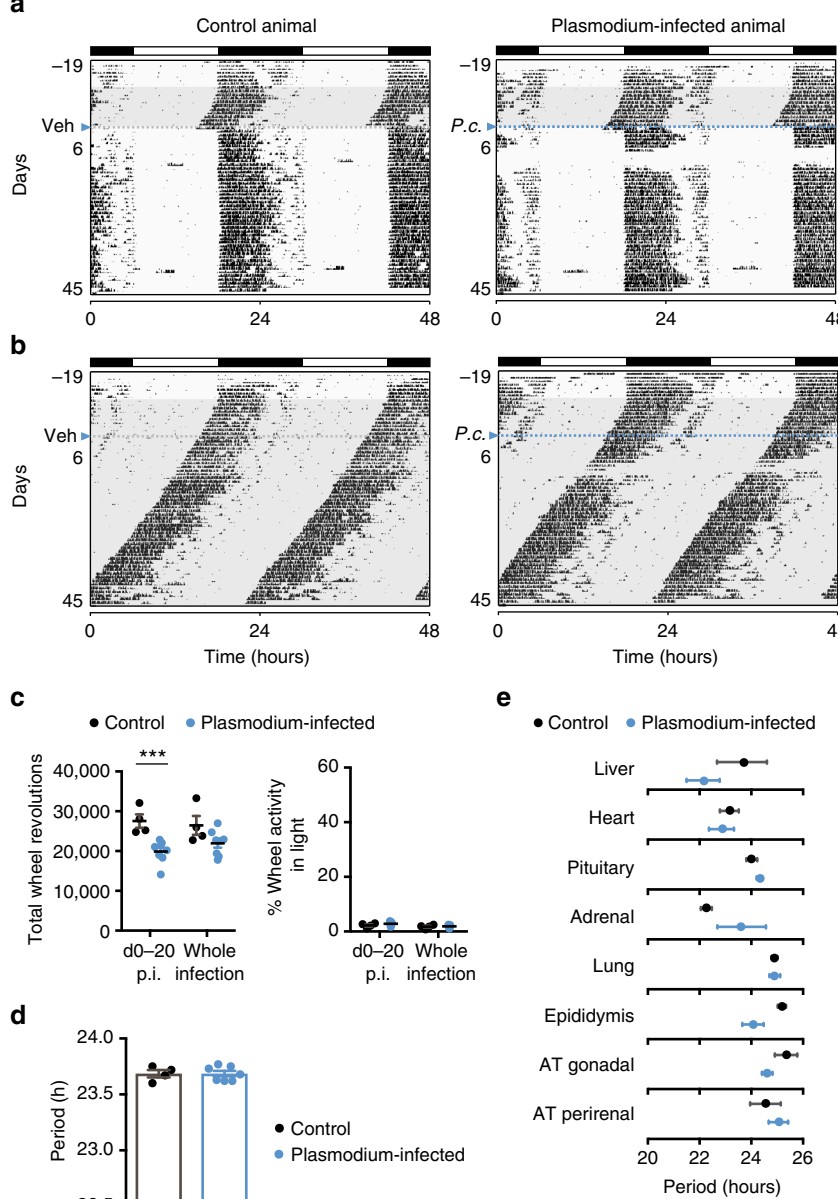

**Fig. 5** Plasmodium infection does not change host's circadian period. **a** Representative actograms of daily wheel-running activity of control and Plasmodium-infected mice (LD). All running-wheel experiments involve seven days on LD12:12 followed by 10 days in DD to confirm that all animals have a normal circadian rhythm, after which animals are either infected or injected with vehicle. Horizontal black and white bars at the top of each actogram represent lights off and on for the initial acclimatization period, respectively. **b** Representative actograms of daily wheel-running activity of control and Plasmodium-infected mice in constant darkness (DD). **c** Total daily and relative rest-period activity levels of control ($n = 4$) and infected ($n = 8$) mice considering the whole infection time and a 20-days post-infection interval. **d** Period of running-wheel activity of control ($n = 4$) and infected ($n = 8$) mice. Error bars show mean ± s.e.m.; *$p < 0.05$, **$p < 0.01$, ***$p < 0.001$ tested with Mann–Whitney test in panels **c** and **d**. **e** PER2::LUC period plot of various tissues harvested from control (black) and Plasmodium-infected (blue) mice. No significance was found, $p > 0.05$, FDR method Benjamini–Hochberg

This could be due to a higher parasite infiltration in the brain parenchyma on day 60 and/or to the fact that the master clock in the SCN has a more robust molecular clock with higher intrinsic resistance to perturbations[22,35–37]. This is likely the reason why the circadian period in the SCN is only 30 min shorter whereas in the highly parasite-infiltrated adipose tissue the period is shortened by 2 h. Curiously, in a previous study using a rat model for sleeping sickness, researchers found a ~30-min period shortening in the pituitary gland using a *Per1-luc* reporter, but not in the SCN[38]. That study did not find behavioral period changes possibly because the rat model was not studied during long-term infection. Here, interestingly, the period change observed in the

mouse SCN ex vivo is similar to the period shortening of mice's running-wheel activity.

Substantial evidence has suggested that inflammation and, in particular, cytokines cause fever, fatigue, and sleep disturbances, which are collectively referred to as sickness behavior syndrome[39]. By injecting TNF, a pro-inflammatory cytokine, the locomotor activity of mice is dramatically decreased[17]. TNF and IL-1β are also both somnogenic, increasing nonrapid eye-movement (NREM) sleep[39,40]. TNF leads to the suppression of the expression of clock downstream genes *Dbp*, *Tef*, and *Hlf* and of the Period genes *Per1*, *Per2*, and *Per3*[17]. So, in general, cytokines[41–43] and even bacterial infection[44] have been shown to

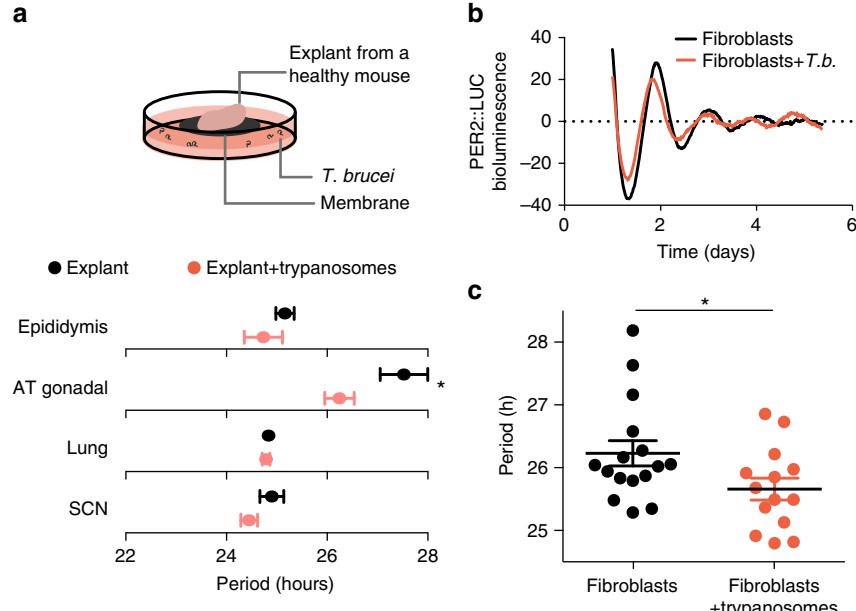

**Fig. 6** *T. brucei* presence is sufficient to shorten the period. **a** Period plot of various tissues harvested from healthy PER2::LUC mice and cultured with medium only (black) or $10^5$ *T. brucei* (orange). Shown are mean period $\pm$ s.d.; *adjusted $p < 0.05$ FDR method Benjamini–Hochberg. **b** Representative records of bioluminescence reporting of circadian expression from PER2::LUC ear fibroblasts co-cultured with $10^5$ *T. brucei* (*T.b.*, orange). **c** Period comparison from PER2::LUC ear fibroblast with medium (black, $n = 16$) and co-cultured $10^5$ *T. brucei* (orange, $n = 14$). Unpaired *t*-test, * $p < 0.05$, from four independent experiments

downregulate circadian clock expression, which may be a common factor to all infections and responsible for the sickness behavior. Although this may explain why Trypanosoma-infected mice run less than healthy control mice, and why both Trypanosoma- and Plasmodium-infected mice show lower transcript levels of clock genes, it cannot explain the differences in the timing and period of clock gene expression. Although TNF injection decreases activity, it does not alter mice circadian rest–activity cycles[17]. Similarly, cytokines damped the oscillations of clock gene expression but always kept the same period[17]. The fact that the period of PER2::LUC Trypanosoma-infected mice did not change on day 6, when there are already very high numbers of circulating inflammatory cells[18,45], and that despite inflammation[46], organs of Plasmodium-infected mice also maintained their period, suggests that period shortening induced during *T. brucei* infection is not due to inflammation. Furthermore, in vitro host cells and explants co-cultured with *T. brucei* showed shorter period. Although many studies show that the outcome of an infection depends on the time of day at which the infection is initiated[44,47,48], *T. brucei* appears be the first infection capable of shortening the period of the host circadian clock.

Here, we show that sleeping sickness infection shortens circadian period and phase advances the host circadian rhythm at the behavior, cellular, and molecular levels. These circadian changes are likely the cause of sleep alterations that are typical of this disease. Further studies are required to identify the systemic signal (secreted by the parasite, or a molecule produced by the host in response to this infection) that could be responsible for such changes in the period of the circadian clock.

## Methods

**Ethics statement**. All animal care and experimental procedures were performed in accordance with (i) University of Texas Southwestern Medical Center (UTSW) IACUC guidelines, approved by the Ethical Review Committee at the University of Southwestern Medical Center and performed under the IACUC-2012-0021 protocol and (ii) European Union (EU) Directive 2010/63/EU, approved by the Animal Ethics Committee of Instituto de Medicina Molecular (IMM) (AEC-2011-

006-LF-TBrucei-IMM), following FELASA guidelines concerning laboratory animal welfare.

**Parasites and culture conditions**. *T. brucei* AnTat 1.1e, a pleomorphic clone, derived from an EATRO1125 strain was originally isolated from the blood of *Tragelaphus scriptus* in Uganda[49]. In all experiments, we used AnTat 1.1e 90-13, a transgenic cell-line encoding the tetracycline repressor and T7 RNA polymerase[50]. For all mice infections, the parasite cryostabilates used were obtained from a previous 5-day infection. Prior to co-culture experiments, bloodstream forms were grown routinely in HMI-11 medium at 37 °C in 5% $CO_2$[51], after which they were transferred to TbM50 medium, described in the circadian bioluminescence section. Parasite numbers were calculated using a Hemocytometer.

**Trypanosoma brucei mouse infection**. The infections of wild-type male C57BL/6J mice, 6–10-week old (UT Southwestern Medical Center Mouse Breeding Core Facility or Charles River Laboratories, France) described in this manuscript, were performed by intraperitoneal (i.p.) injection of 2000 *T. brucei* AnTat 1.1e parasites between ZT9-12 (3-hour interval before lights are off). Prior to infection, *T. brucei* cryostabilates were thawed and parasite viability and numbers were assessed by mobility under a microscope. For circadian luminescence experiments, B6.129S6-$Per2^{tm1Jt}$/J (JAX stock number 006852), which we refer to as PER2::LUC mice in the text, were group housed and maintained in a Specific-Pathogen-Free barrier facility. The facility has standard laboratory conditions: 21–22 °C ambient temperature and a 12 h light/12 h dark cycle (LD12:12). For behavioral and telemetry experiments mice were individually housed. Chow and water were available ad libitum.

**Anemia assessment**. Every Monday and Thursday, a sub-mandibular blood sample was withdrawn between 10:00 and 11:00 AM to measure the hematocrit. Approximately 50 μL of blood was collected into a hematocrit-measuring tube (Hirschmann Laborgerate, Na-heparin 375 IU/capillary). The tube was then centrifuged for 5 min at 12,000 revolutions per minute (rpm) and the hematocrit was measured using a manual reading scale plate.

**Circadian behavioral experiments**. Mice were placed in individual running wheel cages and their activity recorded continuously using the ClockLab data collection system (Actimetrics, Wilmette, IL)[21]. After the first week on a LD (LD12:12) cycle, mice were housed in DD for 10 days. Mice were then infected at ZT9–12 with *T. brucei* either in light when mice were placed back on LD cycles (28 mice) or in the dark when maintained in DD (41 mice) for the remaining of the experiment. On day 21 post-infection, at ZT10–11 (Zeitgeber time, ZT), suramin (20 mg/kg, in 100 μL) was administered i.p. to 14 infected mice in LD cycle and 27 mice in DD, for the long-term infection protocol. For a more graphical explanation of the infection

protocol, please see Fig. 1a–c, e. Animals were checked for health and food status daily with water bottles and bedding being changed every three weeks. Infrared goggles were used during periods of DD. Total daily counts were calculated by averaging the number of wheel revolutions during a 24-h day across the infection, from injection until the end of the experiment. The free-running period in DD was calculated from continuous 10-day periods by using $\chi^2$ periodogram analysis (Clocklab software, Actimetrics, Wilmette, IL).

For the experiment with consecutive LD to DD and back to LD transitions (Fig. 2e), the same initial LD and DD protocol as above was used. On day 60 post-infection, animals were housed in DD for 15 days (DD-1), followed by seven days of LD (LD-1) and new 18 days dark period (DD-2). Phase angles and circular variances of circular plots (Rayleigh plots) were computed using Oriana (Kovach Computing Services, Wales, UK). Multiple t-test comparisons (adjusted for multiple comparisons using the Benjamini–Hochberg procedure for false discovery rate) were used to compare control and infected mice phases across the different dark periods using Graphpad Prism (GraphPad Software, Inc., USA). Behavior was recorded from 6 controls and 14 infected animals.

**Core body temperature measurement**. Twelve mice were surgically implanted with temperature telemeters (telemetry system with a G2 E-Mitter, Mini Mitter Respironics, Bend, OR) inside the peritoneal cavity. Animals were individually housed (without running wheels) and monitored for three weeks during recovery from the surgery. For the animal recordings, cages were placed on top of ER-4000 receivers (Mini Mitter Respironics, Bend, OR) for real-time measurement of body temperature and general cage activity. Baseline recordings correspond to four days in LD12:12 cycles and 10 days in DD, after which seven mice were infected and five were injected with HMI-11 medium vehicle only. For the remaining period of the experiment, the light schedule was LD12:12 cycles. On day 21 post-infection, all animals were injected i.p. with suramin (20 mg/kg) in ~100 µL.

The amplitude of the circadian rhythm was analyzed using the fast Fourier transform (FFT), which estimates the relative power of approximately 24 h period rhythm in comparison with all other periodicities in the time series (spectral analysis, FFT power spectrum, Blackman-Harris windowing) as described previously[52] using ClockLab software (Actimetrics, Wilmette, IL).

**CleverSys behavioral assay**. Six mice were injected i.p. with either HMI-11 trypanosome medium (vehicle) or 2000 T. brucei parasites; 21 days post-infection, animals were injected with suramin (20 mg/kg) in ~100 µL. On days 20, 65, 85, and 95 post-infection, animals were individually housed and their movement recorded for 36 h in LD12:12 using CleverSys software. D1 represents the first 12 h in the dark, L1 the first 12 h in light, and D2 the second night (dark). Data were auto scored by CleverSys and sleep-like behavior manually curated for behavior bouts lasting >300 s. Videos and scoring were visually validated. Statistical analyses were carried out using Graphpad Prism (GraphPad Software, Inc., USA) and all data are represented as mean ± s.e.m. Data distribution was first tested for normality (using the Shapiro test) and data not violating normality were analyzed using two-way mixed design ANOVA test, followed by the Bonferroni post-hoc test (multiple comparisons test). Data that did not conform to normality were analyzed using the Friedman test, followed by Dunn's multiple comparisons test.

**Real-Time qPCR of T. brucei-infected mice**. Animals were housed individually and activity was monitored using a running wheel. Thirty-six animals were either infected or injected with medium and housed in DD. At 20 days post-infection, the locomotor activity of every animal was recorded and analyzed using ClockLab software (Actimetrics, Wilmette, IL) to determine the circadian phase, as previously described[21]. Three animals per phase cluster (6 time points) were euthanized by cervical dislocation, their organs collected, and snap frozen. RNA isolation and real-time PCR were performed as described previously[53,54]. In summary, RNA was extracted using TRIzol™ Reagent (Thermo Fisher Scientific) and reverse transcription was carried out using TaqMan™ Reverse Transcription Reagents (Thermo Fisher Scientific) with random hexamers. Transcript levels were measured in an Applied Biosystems® 7500 Real-Time PCR Systems and normalized to Gapdh. Primer sequences are listed in Supplementary Table 1.

**Real-Time qPCR of P. chabaudi-infected mice**. Thirty-six wild-type male C57BL/6J male mice, 6–10-week old (Charles River Laboratories, France), were either infected by i.p. injection of $1 \times 10^5$ P. chabaudi-infected red blood cells (iRBC) in 200 µL PBS or injected with PBS and used as controls. Parasitemia was assessed daily. On the second peak of parasitemia (chronic infection), animals were housed in DD for 48 h. On day 16 post-infection, three animals per condition in each of 6 time points (every 4 h) were euthanized by cervical dislocation, their organs collected, and snap frozen. RNA was extracted and gene expression quantified by real-time PCR[53], as described in the previous section.

**Histological analysis**. Animals were sacrificed by $CO_2$ narcosis, necropsy was performed, and organs collected and immediately fixed in formalin. The head was decalcified for 3 h using RDO solution. All organs were paraffin-embedded, sectioned at 3 µm, and stained with hematoxilin and eosin (H&E), for routine histopathological analysis. For immunohistochemistry, 3-µm sections were stained for

abundant variant surface glycoprotein (VSG) using a non-purified rabbit serum anti-T. brucei VSG13 antigen (cross-reactive with most VSGs via the C-terminal domain)[4]. Liver and gonadal adipose tissue of Trypanosome- and Plasmodium-infected mice were analyzed for the distribution of inflammatory cell infiltrates, as well as their severity and cell type, and compared with uninfected controls ($n = 5$).

**Circadian bioluminescence experiments**. Trypanosoma-infected and control PER2::LUC (B6.129S6-$Per2^{tm1lt}$/J) mice[27] were euthanized by cervical dislocation between ZT10 and ZT12 on days 6, 20, and 60 post-infection. Two animals were always processed in parallel; one injected with parasites and another with vehicle only. SCN tissues were isolated from 300-µm coronal brain sections, and pituitary, heart, lung, liver, adrenal, adipose tissue, and epididymis were dissected and kept in chilled Hanks' buffered salt solution (Invitrogen) until cultured. All dissected tissues were cultured on Millicell culture membranes (PICMORG50, Millipore) and were placed in 35-mm tissue culture dishes containing 1.2 mL TbM50 medium, composed of 50% (v/v) DMEM media (Mediatech) supplemented with 2 mM L-Glutamine, 25 units/mL penicillin, 25 µg/mL streptomycin (Invitrogen) and 50% (v/v) HMI-11 Trypanosome media and 0.1 mM luciferin potassium salt (L-8240, Biosynth AG). Sealed dishes were placed in a LumiCycle luminometer machine (Actimetrics, Wilmette, IL) and bioluminescence was recorded continuously. Circadian parameters were calculated in the LumiCycle analysis software. Raw data was detrended by subtraction of background bioluminescence based on a running average curve fit to the midline of daily bioluminescence oscillations for all cultures. Circadian parameters of the baseline-subtracted data were determined by a best-fit sine wave analysis within the LumiCycle Analysis software package, with period being defined by how long it takes a full cycle to be completed, phase as the time of the day with maximum PER2 expression in the first cycle, and damping as the damping rate (gradual decrease of amplitude across the days) of such oscillations[55]. Plasmodium-infected PER2::LUC mice were euthanized on day 16 post-infection and organs processed similarly.

Primary ear fibroblasts were isolated from the ear of PER2::LUC mice using 5 mg/mL collagenase I and 0.05% trypsin, followed by a 30-min incubation at 37 °C. For the luminescence assay, cells were plated to confluency in 35 mm dishes with TbM50 medium. Cells were synchronized with a 2-mL TbM50 medium change and half of the fibroblast cultures were incubated with $5 \times 10^4$/mL T. brucei parasites and placed in LumiCycle luminometer with continuous bioluminescence recording for five days. Immortalized PER2::LUC sv/sv fibroblasts[36] were cultured similarly but in 24-well plates. For the healthy PER2::LUC explants co-cultured with parasites, a similar number of parasites were added to the TbM50 recording media.

In the conditioned media experiments, parasites and, in parallel, PER2::LUC sv/sv fibroblasts were grown for three days in the co-culture recording conditions (TbM50 medium, air tight). Pooled parasite cultures were centrifuged at 1800 rpm for 10 min at room temperature, and the supernatant was used as parasite-conditioned media. Supernatant from multiple fibroblast cultures was also pooled and used as fibroblast-conditioned media. Both parasite- and fibroblast-conditioned media were used in 60–40% of fresh TbM50 medium, to ensure synchronization of the receiving fibroblast cultures.

In lysate experiments, parasites were grown in HMI-11 below $1 \times 10^6$/mL density and centrifuged at 3000 rpm for 10 min at 4 °C. Parasite pellets were lysed for 5 min at 37 °C in a hypotonic 10 mM phosphate lysis buffer solution (pH 8.0 containing 0.1 mM TLCK (1-chloro-3-tosylamido-7-amino-2-heptanone; Nα-Tosyl-L-lysine chloromethyl ketone hydrochloride), 1 µg/mL leupeptin, and 1 µg/mL aprotinin protease inhibitors), 33 µL of lysis buffer for every 8 million parasites. Fibroblasts were cultured as described above and synchronized with 100 nM Dexamethasone for 2 h prior to adding the lysates (8 million or 1.6 million) or lysis buffer control for luminescence recording.

**Parasite cell-cycle stage and bloodstream form assessment**. Cultures with a starting inoculum of $5 \times 10^4$/mL T. brucei parasites, in either normal culture conditions (HMI11 medium) or in co-culture recording conditions (TbM50 medium), were monitored for 5 days and viability was recorded. Similar cultures were started using the parasite GFP::PAD1utr reporter cell line, in which a PAD1 3′ UTR sequence is downstream of the coding sequence for Green Fluorescence Protein (GFP), which allows for maximum expression of this protein in stumpy parasite forms. Assessment of bloodstream forms was performed by measuring GFP expression and scoring cell-cycle stage after parasite fixation by slowly adding ethanol to the culture sample to a final concentration of 70%. Fixed trypanosomes were pelleted and DNA was stained using a solution of 0.5 mL PBS/2 mM EDTA containing 10 µg RNAse A and 1 mg propidium iodide (red fluorescence) and incubating samples for 30 min at 37 °C. The percentage of dividing cells was measured in five independent cultures for each of the five days, from a minimum of 30,000 events. The intensity of green and red fluorescence was measured using a FACSCalibur flow cytometer (BD Biosciences) and data were analyzed using FlowJo.

**Purification of soluble variant surface glycoprotein**. For the purification of sVSG from T. brucei, ~$5 \times 10^9$ trypanosomes were grown in culture, harvested, and lysed with 3 mL, at 37 °C, pre-warmed hypotonic lysis buffer (10 mM sodium

phosphate (NaH$_2$PO$_4$), pH 8.0, containing 0.1 mM TLCK, 1 μg/mL leupeptin, and 1 μg/mL aprotinin, Sigma-Aldrich). The supernatant was passed through a column of DE52[56] pre-equilibrated in 10 mM sodium phosphate (pH 8.0). Sample was then concentrated with a Amicon-15 spin concentrator (Millipore); 100 μg of sVSG was used as equivalent of 10$^8$ trypanosomes.

**Data availability**. The authors declare that the data supporting the findings of this study are available within the article and its supplementary information files.

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

## Acknowledgements

We thank Chryshanthi Joseph and Victoria Acosta Rodriguez for help during the implant surgeries of the telemetry system, Lisa Thomas for breeding the PER2::LUC mice and monitoring mice during the end of the phase test experiment, Marleen de Groot and Jennifer Mohawk for training in Clock Lab and LumiCycle software analysis, Vanessa Zuzarte-Luis and Jeremy Stubblefield for help during the 24-h tissue collections, Shin Yamazaki for guidance and discussions of the behavioral experiments, Yoga Chelliah for help in purifying VSG, Yan Li, Iza Kornblum, and Nelly Garduño for assistance during fibroblasts and explant experiments, and Nelly Garduño for monitoring mice infected with *Plasmodium chabaudi*. Research was supported by the Howard Hughes Medical Institute (J.S.T.), by HHMI International Early Career Scientist (55007419) to L.M.F., and by Fundação para a Ciência e Tecnologia (SFRH/BD/51286/2010) to F.R.-F. J.S.T. is an Investigator and F.R.-F. is an Associate in the Howard Hughes Medical Institute.

## Author contributions

F.R.-F., J.S.T., and L.M.F designed the study. F.R.-F. performed the experiments. F.R.-F. and J.S.T. analyzed the data, T.C. performed the histopathological analysis, and C.A., F.R.-F., and R.M.C. performed the CleverSys recordings and analysis. M.S.-V. performed the VSG experiments ex vivo. F.R.-F. wrote the manuscript and all authors contributed to reviewing the manuscript.

## Additional information

**Competing interests:** The authors declare that they have no competing financial interests.

