## [Peer Review File · Nature Communications]

REVIEWERS' COMMENTS:

Reviewer #1 (Remarks to the Author):

At face value, the paper by X et al states the obvious: that the sleep disorder sleeping sickness is in fact a circadian disturbance disorder, likely caused by a factor secreted by the parasite or in response to it.

The authors show convincingly that infection is followed by shortening of the clock (which is separate than the changing of the oscillation amplitude which is often seen under inflammatory conditions). This is correlated with changed temperature rhythms as well as altered activity cycles. It is also overall recapitulated with explanted tissue infected with *T.brucei* ex vivo (where the readouts are RT-PCR for a number of clock related transcripts as well as PER luciferase expression).

Overall it's a tight manuscript that goes from organismal observation all the way down to molecular correlates. Its a pity the authors could not identify a factor(s) mediating this alteration, but that's a tall order for a first manuscript on the topic.

Reviewer #2 (Remarks to the Author):

The authors provide good evidence that sleeping sickness is a consequence of shortening of the circadian period and changes in phase of peripheral tissue clocks due to a secreted factor of the parasite *trypanosoma brucei*. Although the immune system is affected similar to the control infection with *Plasmodium*, the malaria parasite, the experiments show that it is not the changes in the immune system that affects the period. Overall the data support the conclusion that sleeping sickness is a circadian disorder.

Minor points:

line 126 Figure 1 legend: g should be bold.

line 129 D, F and G should be small letters in bold.

line 158, awkward sentence.

line 160, do some animals become arrhythmic or do they die before?

line 164, should read ...may not be due to...

line 177-184, the data are not as clear cut here as they are described in the paragraph. The statements are too strong. I would suggest to say that most of the animals... It is also visible in the graphs that not all animals show a phase advance compared to controls.

line 195, d should be bold.

line 251, d should be bold

line 273, a description of the data for the hypothalamus is missing. The data in supplementary Fig. 4 show that there is no significant difference between infected versus control.

line 299, please give a scale bar in the figure 4d not the magnifications, because the magnification numbers would need to be adapted each time a reproduction is made and the image is not represented 1:1. Same for suppl. figure 1c.

line 342-343, awkward sentence.

line 738, delete one of the ..and...

Suppl fig. 2, Day 85, I would expect here the same difference between D1 and L1 for control animals as in all other days. Probably, the n is too low or do you have another explanation?

Response to Reviewers' Comments:

We greatly appreciate the reviews and comments on our manuscript. In the sections below, each of the reviewers' comments will be in black and our responses will be highlighted in blue.

Reviewer #1 (Remarks to the Author):

At face value, the paper by X et al states the obvious: that the sleep disorder sleeping sickness is in fact a circadian disturbance disorder, likely caused by a factor secreted by the parasite or in response to it.

The authors show convincingly that infection is followed by shortening of the clock (which is separate than the changing of the oscillation amplitude which is often seen under inflammatory conditions). This is correlated with changed temperature rhythms as well as altered activity cycles. It is also overall recapitulated with explanted tissue infected with *T.brucei* ex vivo (where the readouts are RT-PCR for a number of clock related transcripts as well as PER luciferase expression).

Overall it's a tight manuscript that goes from organismal observation all the way down to molecular correlates. Its a pity the authors could not identify a factor(s) mediating this alteration, but that's a tall order for a first manuscript on the topic.

We thank the reviewer for the encouraging comments.

Reviewer #2 (Remarks to the Author):

The authors provide good evidence that sleeping sickness is a consequence of shortening of the circadian period and changes in phase of peripheral tissue clocks due to a secreted factor of the parasite *trypanosoma brucei*. Although the immune system is affected similar to the control infection with *Plasmodium*, the malaria parasite, the experiments show that it is not the changes in the immune system that affects the period. Overall the data support the conclusion that sleeping sickness is a circadian disorder.

Minor points:

line 126 Figure 1 legend: g should be bold.
line 129 D, F and G should be small letters in bold.

We thank the reviewer for helping making the manuscript better.

line 158, awkward sentence.

We rephrased the sentence "This shorter period translates into, after 20 days post-infection, infected mice start to run 12 h earlier than control mice."

To "Due to this period shortening, after 20 days post-infection infected mice start to run 12 h earlier than control mice."

line 160, do some animals become arrhythmic or do they die before?

When measuring this period shortening with running wheels we won't be able to know if animals become arrhythmic because most animals stop running on their wheel a few days before dying, as we tried to document in the actograms in the manuscript by adding "animal not running but alive" and when animal indeed dies as "death". Although it would not change the observation of period shortening, perhaps measuring body temperature in DD would allow to answer that question.

line 164, should read ...may not be due to...
thank you.

line 177-184, the data are not as clear cut here as they are described in the paragraph. The statements are too strong. I would suggest to say that most of the animals... It is also visible in the graphs that not all animals show a phase advance compared to controls.

Yes, this is true there is variability in the degree of animals being affected by the infection. We were using a general "Infected mice" whenever statistics supported the statement and using "some infected mice" to highlight abnormalities/differences that are not statistically sound. But for the interest of transparency we have now changed to "most infected mice".

line 195, d should be bold.
line 251, d should be bold

Altered. Thank you.

line 273, a description of the data for the hypothalamus is missing. The data in supplementary Fig. 4 show that there is no significant difference between infected versus control.

Thank you, the reviewer is correct. We have now included further description of the hypothalamus in the main manuscript and corrected the legend stats in the Supp Fig. The legend did not reflect the *Dbp* gene results that were added later, showing lower amplitude in infected mice (Two-way ANOVA is indeed significant, $p = 0.0054$).

line 299, please give a scale bar in the figure 4d not the magnifications, because the magnification numbers would need to be adapted each time a reproduction is made and the image is not represented 1:1. Same for suppl. figure 1c.

Scale bar now added.

line 342-343, awkward sentence.

We rephrased to:

"Unfortunately, there are technical limitations in order to culture higher numbers of parasites because in high density cultures ($1-3 \times 10^6$ /mL) *T. brucei* differentiate into non-replicative stumpy forms and die soon after ²⁹."

line 738, delete one of the ..and...
Thank you.

Suppl fig. 2, Day 85, I would expect here the same difference between D1 and L1 for control animals as in all other days. Probably, the n is too low or do you have another explanation?

Yes, the n is low and there is some variability. However, in general, we see a tendency for less 'sleep' on D1 than D2 in controls, possibly because animals are more curious/exploring the new cage in the first night. In specific when looking at the data on day 85, what seems to help making the difference between D1 and L1 smaller than other days is the less amount of sleep in L1. This could be because of a loud noise or something that woke up animals that day, that we were not aware of since we were not in the room.